Bile acids induce apoptosis selectively in androgen-dependent and -independent prostate cancer cells

Goldberg Alexander A. 1
Titorenko Vladimir I. 2
Beach Adam 2
Sanderson J. Thomas 1 Thomas.Sanderson@iaf.inrs.ca
1 INRS-Institut Armand-Frappier , Laval, QC , Canada
2 Department of Biology, Concordia University , Montréal, QC , Canada
Moreira Paula
Electronic publication date: 2013 Aug 8
Publication date: 2013
Volume: 1
Electronic Location ID: e122
Received 2013 Mar 20; Accepted 2013 Jul 12
Copyright: © 2013 Goldberg et al.
Copyright year: 2013
Copyright holder: Goldberg et al.
License: This is an open access article distributed under the terms of the Creative Commons Attribution License, which permits unrestricted use, distribution, and reproduction in any medium, provided the original author and source are credited.
License URL: https://creativecommons.org/licenses/by/3.0/

Keywords: Bile acids, Lithocholic acid, Apoptosis, Prostate cancer, Chemotherapy, In vitro, Cell death, Cytotoxicity, Mitochondrial membrane permeability, Extracellular

Funding: Canadian Institutes of Health Research CIHR MOP-115019 This study was supported financially by the Canadian Institutes of Health Research (CIHR MOP-115019). The funders had no role in study design, data collection and analysis, decision to publish, or preparation of the manuscript.

==============================
Prostate cancer is a prevalent age-related disease in North America, accounting for about 15% of all diagnosed cancers. We have previously identified lithocholic acid (LCA) as a potential chemotherapeutic compound that selectively kills neuroblastoma cells while sparing normal human neurons. Now, we report that LCA inhibits the proliferation of androgen-dependent (AD) LNCaP prostate cancer cells and that LCA is the most potent bile acid with respect to inducing apoptosis in LNCaP as well as androgen-independent (AI) PC-3 cells, without killing RWPE-1 immortalized normal prostate epithelial cells. In LNCaP and PC-3 cells, LCA triggered the extrinsic pathway of apoptosis and cell death induced by LCA was partially dependent on the activation of caspase-8 and -3. Moreover, LCA increased cleavage of Bid and Bax, down-regulation of Bcl-2, permeabilization of the mitochondrial outer membrane and activation of caspase-9. The cytotoxic actions of LCA occurred despite the inability of this bile acid to enter the prostate cancer cells with about 98% of the nominal test concentrations present in the extracellular culture medium. With our findings, we provide evidence to support a mechanism of action underlying the broad anticancer activity of LCA in various human tissues.

Introduction

Prostate cancer accounts for approximately 15% of all newly diagnosed cancers and it is the third highest cause of cancer-related deaths in males in the United States (Siegel, Naishadham & Jemal, 2012). Most prostate cancers are initially androgen-dependent (AD) and are generally treated with a combination of radiotherapy, chemical castration, androgen-receptor (AR) antagonists (hydroxyflutamide, bicalutamide), or inhibitors of steroidogenesis (abiraterone). However, a large contingent of AD cancers will progress to become a more aggressive, androgen-independent (AI) form, and less readily treatable, resulting in higher incidences of morbidity and mortality. Furthermore, patients treated with either hydroxyflutamide or bicalutamide are known to suffer from severe side-effects as a result of the anti-androgenic therapy (McLeod, 1997; Wysowski et al., 1993), necessitating the search for natural and more potent anti-cancer compounds with fewer deleterious effects on the human body.

Bile acids are the products of cholesterol catabolism and their main function for the body is the solubilisation of dietary fats and fat-soluble vitamins from the intestinal lumen (Lefebvre et al., 2009; Thomas et al., 2008). More recently, bile acids have been found to have specific regulatory functions, as they interact with a variety of intracellular and extracellular signalling molecules such as the farnesoid X (FXR), vitamin D (VDR), pregnane X (PXR) and G-protein coupled (TGR5) receptors. Numerous studies have shown that bile acids play key homeostatic roles in glucose metabolism, cholesterol and lipid metabolism, xenobiotic detoxification of toxins and lifespan extension (Hylemon et al., 2009; Ramalho et al., 2008; Tiwari & Maiti, 2009; Vallim & Edwards, 2009; Goldberg et al., 2010). Moreover, BAs can be useful small molecules for the treatment of illnesses such as cholestatic liver disease, Alzheimer’s disease, atherosclerosis, obesity and metabolic disorders (Baxter & Webb, 2006; Chen et al., 2011; Hageman et al., 2010; Lefebvre et al., 2009; Lim et al., 2012; Nunes et al., 2012; Peiro-Jordan et al., 2012; Pols et al., 2011; Ramalho et al., 2008; Sola et al., 2003; Thomas et al., 2009; Thomas et al., 2008; Tiwari & Maiti, 2009; Trauner et al., 2010; Viana et al., 2009; Wachs et al., 2005; Zhong, 2010). It has also been reported that bile acids have anti-neoplastic and -carcinogenic properties in a multitude of cancer cell models, such as tamoxifen-resistant breast cancer (Giordano et al., 2011), colon cancer (Katona et al., 2009), prostate cancer (Kim et al., 2006) and neuroblastoma (Goldberg et al., 2011) cells.

We have previously reported that LCA delays chronological aging of the budding yeast, S. cerevisiae, independent of AMP-activated protein kinase/target of rapamycin (AMPK/TOR) and cAMP/Protein Kinase A (PKA) signalling. LCA alters the age-related dynamics of metabolomic processes in yeast such as respiration and reactive oxygen species production in the mitochondria, and lipid and trehalose accumulation, and modulates stress response pathways (Goldberg et al., 2010). Moreover, we have shown that LCA can kill human neuroblastoma cells, while sparing normal human primary neurons. LCA selectively initiates an extrinsic apoptotic programme of cell death in neuroblastoma cells, thus recruiting and activating the initiator caspase-8, inducing mitochondrial outer membrane permeabilization (MOMP), mitochondrial fragmentation, and ultimately activation of the downstream proteases caspase-9 and -3 (Goldberg et al., 2011). Here we report that bile acids can inhibit dihydrotestosterone (DHT)-induced cell proliferation, and kill both AD and AI prostate cancer cells in a caspase-3 dependent manner, by eliciting the intrinsic and extrinsic pathways of apoptosis. The mechanistic studies we present will further our understanding of the potential of bile acids to act as chemotherapeutic agents against prostate cancer.

Materials and Methods

Cell lines and reagents

LNCaP, PC-3 and RWPE-1 cell lines were purchased from ATCC (Manassas, VA). LNCaP and RWPE-1 cells were grown in RPMI 1640 supplemented with 10% fetal bovine serum or 2% dextran-coated charcoal-stripped FBS, 2 mM L-glutamine, 1% HEPES, 1% sodium-pyruvate and 10 ml/L of 100 × antibiotic-antimycotic solution. PC-3 cells were grown in a 1:1 mixture of DMEM and Ham’s F12 Nutrient Mixture with either 10% fetal bovine serum or 2% dextran-coated charcoal-stripped FBS, 2 mM L-glutamine and 10 ml/L of 100 × antibiotic-antimycotic solution (Sigma-Aldrich, St. Louis, MO). Cells were maintained in a humidified atmosphere (5% CO2) at 37°C. Lithocholic acid (LCA), deoxycholic acid (DCA), chenodeoxycholic acid (CDCA), ursodeoxycholic acid (UDCA), hyodeoxycholic acid (HDCA) and cholic acid (CA) were purchased from Sigma-Aldrich and dissolved in DMSO to make 500 mM stock solutions. Dihydrotestosterone (DHT; Steraloids Inc., Newport, RI) was dissolved in DMSO to make 100 mM stock solutions. The final concentration of DMSO in culture medium was not greater than 0.2%. The selective caspase substrates Ac-DEVD-AFC, Ac-IETD-AFC and Ac-LEHD-AFC were purchased from Enzo Life Sciences (Farmingdale, NY) and dissolved in DMSO to make 20 mM stock solutions. Caspase inhibitors z-DEVD-fmk and z-IETD-fmk (BD Biosciences, Franklin Lakes, NJ) were dissolved in 100% DMSO to produce 10 mM stock solutions. All primary antibodies were purchased from Cell Signaling (Beverly, MA). LNCaP and PC-3 cells were exposed to 1000-fold dilutions of the appropriate stock solutions of bile acids, DHT and/or caspase inhibitors in their respective experimental culture media. Control cells were exposed to 0.1% or 0.2% DMSO for single or co-exposure experiments, respectively.

LNCaP cell proliferation

LNCaP cells were seeded in 16-well E-plates (Roche Diagnostics, Laval, QC) at a density of 25 × 103 cells per 200 µl medium containing 2% stripped FBS/well. After 24 h, DHT was added at a concentration (0.1 nM) that stimulated optimal growth rate (without surpassing confluence) in culture over a 72 h period along with various concentrations of LCA or DMSO vehicle. Then cell proliferation was determined quantitatively and in real-time over a period of 72 h by measuring changes in impedance detected by the gold electrode-microarrays at the bottom of each of the 16 wells of the E-plate.

Apoptosis, necrosis and mitochondrial membrane potential

For apoptotic and necrotic cell death measurements, LNCaP and PC-3 cells were seeded at densities of 1 × 105 and 0.5 × 105 cells/ml, respectively, in 24-well plates in 2% stripped-FBS. Cells were then treated with several concentrations of bile acids in the presence (LNCaP) or absence (PC-3) of 0.1 nM DHT. After 48 h, Hoechst 33342 (Sigma-Aldrich) and propidium iodide (PI; Invitrogen, Carlsbad, CA) were each added at a concentration of 1 µg/ml per well. After a 15 min incubation at 37°C, cells were observed and counted under a Nikon Eclipse (TE-2000U) inverted fluorescent microscope at 20× magnification. Hoechst-positive and PI-positive cells were made visible using filter cubes with excitation wavelengths of 330–380 nm and 532–587 nm, respectively. To measure mitochondrial membrane potential (MMP), cells were treated with LCA for 1, 4 or 8 h, and tetramethylrhodamine ethyl ester (TMRE) was added to each well at a final concentration of 50 nM. TMRE is a cell permeable, positively charged dye that accumulates in active negatively charged mitochondria. In inactive or depolarized mitochondria, membranes have decreased potential and fail to sequester TMRE. After a 15 min incubation at 37°C, cells were observed under an inverted fluorescent microscope using a filter cube with excitation wavelengths of 532–587 nm. The photos were then analyzed using ImageJ image processing software (Schneider, Rasband & Eliceiri, 2012).

Caspase activity assays

PC-3 and LNCaP cells were seeded in 6-well plates at densities of 400000 or 750000 cells per well, respectively, in 0.5 ml culture medium containing 2% stripped FBS and 24 h later they were exposed to various concentrations of LCA in fresh medium for another 24 h. Proteins were then extracted from harvested cells using 1× RIPA buffer (Millipore, Billerica, MA) containing 1× protease inhibitor cocktail, centrifuged at 13,000 g for 5 min at 4°C to remove cell debris, and frozen at −80°C overnight. Protein concentrations were then quantified using a BCA protein detection kit (Thermo Scientific, Waltham, MA). Caspase activities were determined using fluorogenic caspase substrates selective for either caspase-3 (5 µM Ac-DEVD-AFC), caspase-9 (10 µM Ac-LEHD-AFC) or caspase-8 (10 µM Ac-IETD-AFC) in 10 µg of extracted protein suspended in caspase reaction buffer (20 mM PIPES at pH 7.2, 30 mM NaCl, 10 mM DTT, 1 mM EDTA, 0.1% CHAPS, 10% sucrose). The time-dependent release of 7-amino-4-trifluoromethyl coumarin (AFC) was measured using a SpectroMax M5 microplate reader (Molecular Devices, Sunnydale, CA) at an excitation wavelength of 400 nm and an emission wavelength of 505 nm. Measurements were recorded at 2 min intervals for 90 min. A standard curve of AFC fluorescence was used to calculate the amount of AFC released (in picomoles) during each reaction.

SDS-PAGE and Western Blot

Crude protein extracts (50 µg) were resolved by electrophoresis using 10% sodium dodecyl sulfate-polyacrylamide gels and then transferred to a PVDF Immobilon-P membrane (Bio-Rad, Mississauga, ON). Blots were blocked using 5% milk powder (Selection brand, Marché Jean-Talon, Montréal, QC) and incubated with antibodies as follows: 1:250 dilution for anti-caspase-3, 1:1000 for anti-cleaved PARP, 1:1000 for anti-Bcl-2, 1:1000 for anti-Bax, and 1:1000 for anti-Bid. Immunoreactive proteins were exposed to anti-rabbit horseradish peroxidise-conjugated secondary antibodies (Millipore) that were diluted 1:5000. Antigen-antibody complexes were detected using Immobilon ECL Western Chemiluminescent HRP Substrate (Millipore) and recorded with a Versadoc imaging system (Bio-Rad). Total protein content per well was determined using 1 × Amido Black staining solution (Sigma-Aldrich).

Mass spectrometry

Mass spectrometry-based analysis of LCA and UDCA was performed as previously reported (Bourque & Titorenko, 2009). In brief, lipids were extracted by a modified Bligh and Dyer method (Bourque & Titorenko, 2009) from cells pelleted by centrifugation for 5 min at 16,000 × g at 4°C and from the supernatant of cultural medium. The extracted lipids were dried under nitrogen and resuspended in chloroform. Immediately prior to injection the extracted lipids were combined with a 2:1 methanol:chloroform mixture supplemented with 0.1% (v/v) ammonium hydroxide. The sample was injected directly into a Thermo Orbitrap Velos equipped with a HESI-II ion source (Thermo Scientific, Waltham, MA, USA) at a flow rate of 5 µl/min. Spectra were obtained in negative-ion mode. The source voltage was set to 4.0 kV, a capillary temperature of 275°C, a sheath gas flow of 5 (arbitrary units) and an auxiliary gas flow of 1 (arbitrary units). Acquired spectra were exported from Xcalibur software (Thermo Scientific) and then deconvoluted and deisotoped using Excel macros.

Statistical analysis

All experiments were performed in at least triplicate using cells after various passages and the data are presented as mean ± SEM. Statistically significant differences (p < 0.05) between various treatments and untreated cells were determined using a two-tailed Student t-test with Bonferroni correction for multiple comparisons. IC50 values for inhibition of cell viability were calculated using a sigmoidal curve-fitting model of log-inhibitor concentration versus normalized inhibition response, with variable slope (GraphPad Prism v5.03, GraphPad Software, San Diego, CA).

Results

Bile acids inhibit proliferation and induce cell death in LNCaP and PC-3 cells

A 48 h treatment with LCA significantly decreased the number of intact LNCaP and PC-3 cells, with IC50 values of 40.5 ± 0.07 µM and 74.9 ± 0.25 µM, respectively, without decreasing the viability of non-tumorigenic RWPE-1 cells (Fig. 1A). The hydrophobic bile acids DCA and CDCA were less cytotoxic than LCA, decreasing cell viability at concentrations above 100 µM in LNCaP and PC-3 cells (Figs. 1B and 1C). Relatively hydrophilic bile acids, such as HDCA and UDCA, decreased the number of intact cells at concentrations above 300 µM in either cell line, whereas CA was not cytotoxic at concentrations as high as 500 µM.

Figure 1 Bile acids inhibit proliferation and induce apoptosis in androgen-dependent LNCaP and -independent PC-3 prostate cancer cells.

(A) Percentage of intact LNCaP, PC-3 and RWPE-1 cells that did not have fragmented nuclei (apoptotic), condensed chromatin (apoptotic), or propidium iodide staining (necrotic) was calculated 48 h after treatment with 50 or 75 µM of lithocholic acid (LCA). The percentage of intact LNCaP cells (B) and PC-3 cells (C) was calculated 48 h after treatment with increasing concentrations (10–500 µM) of lithocholic (LCA, •), deoxycholic (DCA, ■), chenodeoxycholic (CDCA, □), hyodeoxycholic (HDCA, ▴), ursodeoxycholic (UDCA, Δ) or cholic (CA, ∘) acid. (D) Relative androgen-dependent growth rates of LNCaP cells grown in stripped RPMI 1640 medium without phenol-red and co-treated with 0.1 nM DHT and increasing concentrations (1–25 µM) of LCA. Data are presented as means ± SEM (n = 3–5).

In addition to LCA-mediated inhibition of cell viability, we assessed the ability of lower concentrations of LCA to inhibit the AD proliferation of AR positive LNCaP prostate cancer cells when stimulated with DHT. Indeed, LCA decreased the proliferation of androgen-stimulated LNCaP cells in a concentration-dependent manner with an IC50 of 8.5 µM ± 1.9 (Fig. 1D).

LCA induces a caspase-3-dependent apoptotic programme

To determine whether the caspases play a role in bile acid-induced prostate cancer cell death, we determined the effects of LCA on caspase-3 activity in AD LNCaP and AI PC-3 cells. LNCaP and PC-3 cells exposed to sub-cytotoxic and cytotoxic concentrations of LCA for 24 h contained increased levels of the cleaved and active 17 and 20 kDa subunits of the 34 KDa caspase-3 zymogen (Fig. 2A). In concordance with this observation, the catalytic activity of caspase-3 was also increased after exposure to (sub)cytotoxic concentrations of LCA (Fig. 2B). Also, levels of the 89 kDa fragment of poly ADP ribose polymerase (PARP), an endogenous substrate of caspase-3 usually cleaved during apoptosis, were significantly elevated in LNCaP cells, but not in PC-3 cells (Fig. 2C). Moreover, a cell permeable inhibitor of caspase-3, z-DEVD-fmk, partially inhibited LCA-induced cell death in both cell lines (Fig. 2D).

Figure 2 LCA-induced cell death is a caspase-3-dependent process.

Cleavage of caspase-3 protein was assessed by western blot (A) and catalytic activity (B) was measured by cleavage of the fluorogenic substrate Ac-DEVD-AFC in response to a 24 h treatment of LNCaP cells and PC-3 cells with increasing concentrations (25–75 µM) of LCA. (C) Cleavage of PARP after 24 h exposure of LNCaP cells to increasing concentrations (25–75 µM) of LCA. (D) Inhibition of cell death after a 24 h co-exposure of LNCaP (40 µM) or PC-3 (50 µM) cells to LCA and 10 µM of the membrane permeable caspase-3 inhibitor z-DEVD-fmk. In (B) and (D) responses are presented as means ± SEM (n = 3–5); ∗p < 0.05; ∗∗∗p < 0.001.

LCA does not accumulate inside LNCaP or PC-3 cells

To determine the extent to which LCA was able to enter human prostate cancer cells, we determined the intra/extra cellular distribution of LCA under our experimental cell culture conditions. LNCaP and PC-3 cells did not accumulate LCA, with as much as 98% of the nominal LCA concentrations present in the extracellular medium of LNCaP and PC-3 cultures after 24 h (Table 1). Also, neither cell line was able to accumulate the relatively hydrophilic bile acid, UDCA, when treated with concentrations as high as 75 µM for 24 h (Table 1).

Table 1 Extra/intracellular distribution of LCA and UDCA in LNCaP and PC-3 human prostate cancer cells in culture.

Cells were separated from cultural media and the compounds were extracted and their concentrations measured by mass spectrometry as described in Materials and Methods. Percentages are presented as means ± SD of three independent experiments.

Cell line	Compound	Concentration (µM)	% of compound recovered	
			Medium	Cells	
LNCaP	LCA	25	97.28 ± 1.10	2.72 ± 1.10	
	50	94.15 ± 2.45	5.85 ± 2.45	
UDCA	25	90.40 ± 3.86	9.60 ± 3.86	
	50	95.77 ± 0.10	4.23 ± 0.10	
PC-3	LCA	50	97.91 ± 0.05	2.09 ± 0.05	
	75	97.61 ± 1.93	2.39 ± 1.93	
UDCA	50	97.56 ± 0.17	2.44 ± 0.17	
	75	97.57 ± 0.67	2.43 ± 0.67	

LCA activates extrinsic and intrinsic pathways of apoptosis in human prostate cancer cell lines

The inability of LCA to significantly accumulate inside prostate cancer cells led us to explore if LCA-induced cell death may occur through activation of the extrinsic pathway of apoptosis. We found increased levels of active caspase-8 in extracts of both LNCaP and PC-3 cells treated with increasing concentrations of LCA (Fig. 3A). LCA-induced cell toxicity was also alleviated in the presence of the caspase-8 inhibitor, z-IETD-fmk (Fig. 3B). Moreover, we found statistically significant increases of caspase-9 activity in both cell lines (Fig. 4A) in addition to cleavage of pro-apoptotic Bcl-2 related proteins Bax and Bid (Fig. 4B). However, we found decreased levels of Bcl-2 in PC-3 cells only (Fig. 4B). We observed only a slight decrease in MMP in LNCaP cells after 8 h of exposure to LCA, but in PC-3 cells we observed a marked decrease in MMP as early as after 1 h of treatment with LCA (Fig. 4C).

Figure 3 LCA activates the extrinsic pathway of apoptosis in androgen-dependent and -independent prostate cancer cells.

(A) Activity of caspase-8 was measured by cleavage of the fluorogenic substrate Ac-IETD-AFC after 24 h of treatment of LNCaP and PC-3 cells with increasing concentrations (25–75 µM) of LCA. (B) Inhibition of cell death after a 24 h co-exposure of LNCaP (40 µM) or PC-3 (50 µM) cells to LCA and 10 µM of the membrane permeable caspase-8 inhibitor z-IETD-fmk. Activities are presented as means ± SEM (n = 3–5); ∗p < 0.05; ∗∗p < 0.01.

Figure 4 LCA activates the intrinsic pathway of apoptosis in androgen-dependent and -independent prostate cancer cells.

(A) Activity of caspase-9 was measured by cleavage of the fluorogenic substrate Ac-LEHD-AFC after a 24 h treatment of LNCaP and PC-3 cells with increasing concentrations (25–75 µM) of LCA. (B) Expression levels of Bcl-2 and cleavage of Bax and Bid after a 24 h exposure of LNCaP and PC-3 cells to increasing concentrations (25–75 µM) of LCA. (C) Mitochondrial membrane permeability was measured using TMRE in LNCaP and PC-3 cells treated with 50 and 75 µM LCA, respectively. In (A) and (C) responses are presented as means ± SEM (n = 3–5); ∗p < 0.05; ∗∗p < 0.01.

Discussion

We have previously shown that LCA can selectively kill human neuroblastoma cells at concentrations non-toxic to normal human primary neurons (Goldberg et al., 2011). In the present study, we provide evidence that LCA also possesses selective anticancer properties against cultured AD and AI prostate cancer cells, whilst not affecting the viability of normal epithelial prostate cells.

LCA activates a caspase-dependent mode of apoptosis in AD and AI prostate cancer cells

We have shown that LCA kills LNCaP and PC-3 cells in a caspase-dependent manner by activating the intrinsic and extrinsic pathways of apoptosis. In both prostate cancer cell types, cell death induced by LCA appears to be at least partly dependent on the activity of initiator caspase-8. Our observation is similar to those made in studies where treatment of hepatocytes and colon cancer cells with bile acids resulted in a TGR5-dependent increase in levels of CD95/Fas death receptor in the plasma membrane, facilitating the activation of caspase-8 and its downstream apoptotic machinery (Katona et al., 2009; Yang et al., 2007). TGR5 is a cell surface membrane-bound metabotropic G-protein-coupled receptor that is highly conserved among species and is found predominantly in gall bladder and intestinal epithelium (Foord et al., 2005; Tiwari & Maiti, 2009). Although TGR5 mRNA has been found in prostate cells (Kawamata et al., 2003), its function in this tissue has yet to be established. LCA is a potent natural agonist of TGR5 and, upon direct binding to the receptor, activates a cAMP/PKA signalling cascade, resulting in the modification of the oxidation-reduction processes in the mitochondria with a resultant increase in the generation of reactive oxygen species, thereby promoting the vesicle-mediated trafficking of CD95/Fas from the Golgi apparatus to the plasma membrane (Hylemon et al., 2009; Katona et al., 2009; Merrill et al., 2011; Pols et al., 2011; Sodeman et al., 2000; Thomas et al., 2008; Wachs et al., 2005). Interaction between LCA and TGR5 may also stimulate the phosphorylation of c-Jun-N terminal kinase (JNK) through activation of the MEKKK1/2/3-MKK4/7 pathway, resulting in the release of pro-caspase-8 from JNK and enabling its recruitment to CD95/Fas (Yang et al., 2007). That LCA does not accumulate inside either LNCaP or PC-3 cells implies that LCA interacts directly with a cell surface receptor in order to activate the extrinsic pathway of apoptosis, and that this identity of this receptor is likely to be TGR5 as it is the only known cell surface receptor that binds bile acids, a hypothesis we are currently investigating in detail.

Our observation that Bid is cleaved after treatment with LCA suggests that once caspase-8 is activated it would continue to cleave Bid, thereby initiating the intrinsic pathway of apoptosis. In fact, treatment of neuroblastoma cells with LCA resulted in MOMP, which allowed the exit of cytochrome c, formation of the apoptosome and ultimately activation of caspase-9 (Goldberg et al., 2011). In the present study, we show that LCA induces Bax cleavage, suggesting an induction of MOMP and mitochondrial fragmentation, resulting in the observed activation of caspase-9 in both cell lines. Moreover, we show that LCA causes loss of MMP in PC-3 cells as soon as 1 h after treatment, showing that LCA induces MOMP in at least one type of prostate cancer cell, in addition to neuroblastoma cells, and that the induction of MOMP is an early-stage event in the induction of apoptosis in these cells. We did not observe a statistically significant decrease in MMP in LNCaP cells up to 8 h after exposure to LCA, indicating that the onset of MOMP in these cells is a later-stage event that would appear to occur after activation of caspase-8. Therefore, it is possible that LCA may transmit a MOMP-activating signal through interaction with a cell surface receptor that requires numerous steps in order to target the mitochondria and activate the intrinsic pathway of apoptosis. Additionally, the reduction in the levels of Bcl-2 observed only in PC-3 cells may suggest a larger role for Bcl-2 in promoting the intrinsic pathway of apoptosis in these cells, and may help explain why LCA increased MOMP much earlier in PC-3 cells than in LNCaP cells.

We saw increases in the activity of caspase-3 in each cell line treated with LCA, and cell death induced by LCA was only partially dependent on this main effector caspase. Previous studies have shown that the cleaved form of Bax can significantly sensitize cells to stress-induced apoptosis by releasing cytochrome c, apoptosis-inducing factor and endonuclease G from the mitochondria (Cao, Deng & May, 2003; Gao & Dou, 2000; Toyota et al., 2003; Wood & Newcomb, 2000; Cabon et al., 2012; Moubarak et al., 2007; Whiteman et al., 2007). Paired with our observation that inhibition of caspase-8 did not completely abrogate LCA-induced cell death, it is likely that LCA induces both caspase-dependent and -independent modes of apoptosis in prostate cancer cells.

Pharmacophore modeling of the anticancer activity of bile acids

We tested the ability of a wide range of bile acids to induce cell death in AD (LNCaP) and AI (PC-3) prostate cancer cells and found that LCA was the most effective bile acid, while two other hydrophobic bile acids, DCA and CDCA, were moderately toxic to these cells. Addition of alpha-oriented hydroxyl groups at the 7- or 12-positions (CDCA and DCA, respectively) or at the 6-position (HDCA) significantly reduced the cytotoxicity of the bile acid structure. Moreover, addition of beta-oriented hydroxyl groups to the molecule at the 12-position (UDCA) further reduced the toxicity of the bile acid in each cell line, whereas the addition of two alpha-oriented hydroxyl groups to both the 7- and 12-positions rendered the resultant bile acids non-cytotoxic. Therefore, reduction of the hydrophobicity of the alpha- or beta-faces of the steroid backbone is sufficient to negate the toxicity of the molecule. Such a relationship between bile acid hydrophobicity and potency, where the hydrophobicity is directly correlated with the biological activity of the molecule, has been described in two other contexts: (1) TGR5 receptor activation in a reporter system (Kawamata et al., 2003), and (2) extension of lifespan of chronologically aging yeast (Goldberg et al., 2010). In support of a possible involvement of TGR5 in bile acid-induced cell death, we found a highly significant correlation between the potency of each cytotoxic bile acid in LNCaP cells and its reported EC50 for induction of TGR5 receptor-mediated luciferase activity (r = 0.96; n = 4; p < 0.001) in transiently transfected Chinese Ovarian Hamster cells (Sato et al., 2008). The inability of LCA and UDCA to enter LNCaP or PC-3 cells would suggest they would have equal opportunity to interact with a cell surface membrane receptor in order to induce apoptosis, yet LCA was far more potent than UDCA, supporting the notion that a specific interaction between LCA and a cell surface receptor, possibly TGR5, could be the event responsible for the apoptotic death of prostate cancer cells. With respect to the lesser toxic bile acids, our results are consistent with a previous study of bile acids in PC-3 cells where concentrations of CDCA and UDCA as high as 100 µM did not result in significant cell death (Choi et al., 2003).

LCA inhibits proliferation of AD prostate cancer cells

In our study, LCA inhibits the proliferation of DHT-stimulated LNCaP cells, yet it is unlikely that LCA directly antagonizes the AR, because it does not accumulate inside LNCaP cells. Instead, it is possible that the inhibition of LNCaP cell growth is related to the ability of LCA to interact with cell surface receptors, such as TGR5, which can activate JNK, thereby antagonizing the NFκB pro-survival pathway (Yang et al., 2007), or any other cell surface receptor capable of inhibiting androgen-independent processes related to proliferation of these cells. It has previously been reported that LCA can directly bind to two key negative regulators of p53, MDM2 and MDM4 (Vogel et al., 2012). However, it is unlikely that LCA directly inhibits either of these cytoplasmic proteins directly due to its inability to enter prostate cancer cells. It is still possible that LCA might upregulate p53 expression via a Mnt-Max to Myc-Max switch in nuclear binding of E-box sequences in these cells (Yang et al., 2009), but our results imply that this would be more likely the result of an upstream event, such as the activation of a cell surface receptor.

Broad anticancer activity of LCA toward various cancerous tissues

Our findings in this report share several similarities with those we described previously in neuroblastoma cells treated with LCA (Goldberg et al., 2011): (1) LCA elicits apoptosis in a caspase-3 dependent manner, (2) LCA activates both extrinsic and intrinsic pathways of apoptosis and (3) LCA enters neither neuroblastoma nor prostate cancer cells. It is then probable that these cancerous tissues share a common target, which is most likely localized at the surface of the plasma membrane and is either activated or deactivated by LCA in order to elicit apoptosis. Additionally, the concentrations used to kill prostate cancer was found to be in a similar range to that of neuroblastoma cells (between 25 and 100 µM), and these concentrations were found to be non-toxic to both normal human primary neurons and non-tumourigenic immortalised prostate cells. We have also found that concentrations well below those needed to induce apoptosis (2.5–10 µM) can inhibit the proliferation of AD prostate cancer cells. Though ingesting amounts of LCA in order to reach plasma concentrations as high as these might be lethal, methods of employing LCA in a more targeted manner, using nanoparticle-encapsulation techniques (Makadia & Siegel, 2011) or delivery via infection with the bacteria Listeria monocytogenes (Quispe-Tintaya et al., 2013), allowing for its accumulation in only immune-compromised metastatic tissues without killing the surrounding tissues, could be successfully developed to employ LCA as an anti-cancer compound. It is then pertinent to understand the exact molecular mechanism of cell death instigated by LCA, to develop novel strategies to enhance the ability of LCA or newly designed compounds to trigger (the) LCA-mediated anticancer pathway(s), as well as to validate these strategies using in vivo models of neuroblastoma, prostate and other cancers.

Additional Information and Declarations

Competing Interests

Author Contributions

None of the authors have competing interests.Thomas Sanderson is an Academic Editor for PeerJ, but was not involved in the review process of this manuscript.

Alexander A. Goldberg conceived and designed the experiments, performed the experiments, analyzed the data, contributed reagents/materials/analysis tools, wrote the paper.

Vladimir I. Titorenko analyzed the data, contributed reagents/materials/analysis tools, wrote the paper.

Adam Beach performed the experiments, analyzed the data, contributed reagents/materials/analysis tools.

J. Thomas Sanderson conceived and designed the experiments, analyzed the data, contributed reagents/materials/analysis tools, wrote the paper.

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
