# Peer review of "Bile acids induce apoptosis selectively in androgen-dependent and -independent prostate cancer cells"

_PeerJ, doi:10.7717/peerj.122_

## Round 0.1 · original submission · Major Revisions

Dear Dr. Sanderson:

Thank you for submitting your manuscript "Bile acids induce apoptosis in androgen-dependent and -independent prostate cancer cells" to PeerJ. We have now had a detailed assessment of your paper by our reviewers and their comments are below for your information.

·

Basic reporting

No comments

Experimental design

No comments

Validity of the findings

Using androgen-dependent and -independent prostate cancer cells, Goldberg and collaborators observed that bile acids, particularly lithocholic acid are able to prevent dihydrotestosterone-promoted cell proliferation and induce apoptotic cell death through the induction of both intrinsic and extrinsic pathways. Despite the potential relevance of this study for the oncobiology/oncotherapy field, some points need to be addressed:
Major points:
1. Mechanistically, this is a poor study since the molecular basis behind the apoptotic effects of bile acids was not too much explored. What is happening upstream caspases activation? The authors are encouraged to improve the mechanism(s) underlying bile acids- triggered apoptosis in prostate cancer cell lines. For instance, the evaluation of some mitochondrial function, morphology and metabolic parameters will improve the manuscript;
2. Concerning Western blot data, the authors should present the statistic and not only a representative image. Also an improved representative Figure 2D is required.

Additional comments

Minor points:
1. Page 5, lines 85 to 88 - the font used is different;
2. A thorough review of the manuscript for abbreviations and typographical errors is required;

Reviewer 2 ·

Basic reporting

In the manuscript “Bile acids induce apoptosis in androgen-dependent and -independent prostate cancer cells” the authors hypothesized that bile acids can induce apoptosis in androgen-dependent and –independent prostate cancer cells. The manuscript is well written and the results are relevant for the hypotheses. Nevertheless, there are several drawbacks that should be at least discussed.

a) The authors did not test the bile acids effects in a noncancerous prostate epithelial cell line such as PNT1A. This deserves clarification since these cells should be the real control of the experiments. Moreover, some speculations should also be avoided. For instance “the potential of bile acids to act as chemotherapeutic agents against prostate cancer” can only be evaluated after studying their effect in noncancerous cells.

b) The bile acids concentrations used should also be discussed. Which concentrations can be considered physiologic? How the concentrations now detected as able to induce apoptosis can affect noncancerous cells?

c) The hypothesis is clear and the approach is sufficient to study the hypothesis. If the authors proposed to compare androgen-dependent and –independent prostate cancer cells then the results should also be presented in that way. Therefore, I would recommend the authors to present new figures comparing LNCaP and PC3 in the same graphs instead of the approach now presented. In figures 2, 3 and 4 the multipanels can be merged. For instance, figure 2 panel B and C should be merged, as well as panel E and F.

Experimental design

a) The research question is pertinent and the paper has merit. The experimental design is correct given the authors hypothesis that bile acids can induce apoptosis in androgen-dependent and –independent prostate cancer cells. However the author’s conclusions are confined to compare only these two cancer situations and the aggressiveness of prostate cancer which is related to androgen response. Speculation should be limited since noncancerous cells were not used in this study.

b) The methods section is poor. The mass spectroscopy methods are not described. Please correct.

Validity of the findings

a) Why is the N between 3 and 5? Moreover, it is not well explained if the authors used different cell passages or only replications.

b) Speculation must be clearly identified as such. The hypothesis is only related to different cancerous cell lines.

c) Please provide explanation to the use of 0.1 nM of DHT.

d) Page 9, line 247 “The inability of LCA and UCDA to enter LNCaP .. death of prostate cancer cells” – This is not clear. High-affinity interaction between LCA and TGR5 is difficult to detect from these results.

e) page 9, line 251 “… are consistent with a previous study…” lacks reference.

f) page 10, line 255 “… it is unlikely that LCA directly antagonizes the AR, because it does not accumulate…”. Please explain. If LCA can cross the membrane then it can directly act on AR.

Additional comments

This paper has merit however it presents several pitfalls. The scientific outcome of the manuscript can be improved after a careful revision. The bile acids effect on noncancerous cells should be at least discussed.

---

## Round 0.2 · Major Revisions

Please make the necessary changes according to the reviewer's comments.

Reviewer 1 ·

Basic reporting

The revised form of the manuscript “Bile acids induce apoptosis in androgen-dependent and -independent prostate cancer cells” has improved the significance of the work. Nevertheless, it is not acceptable in his current form since Mass Spectroscopy methods are still lacking! As previously said, is it crucial to include that information. Moreover, TGR5 is presented as a target for LCA without exploring or even explaining the basis for such assumption.

Experimental design

a) The methods section still presents problems! Please correct. All methods must be described.

Validity of the findings

a) 2 points previously raised concerning TGR5 were not clearly answered. The authors state that LCA interaction with cell surface receptors is likely to occur with TGR5 as a “…scientifically justified target” although “…we do not know if LNCaP and PC-3 cells express TGR5”. Thus, it is imperative to justify carefully this issue, or quantify TGR5 under these circumstances and present those results.

b) Again: “… they would have equal opportunity to interact with a cell surface membrane receptor such as TGR5 (…) supporting the notion that a high-affinity interaction between LCA and TGR5….”. This high-affinity interaction is not proved or supported. Such conclusion needs strong scientific support rather than a hypothesis. It is OK to suggest the interaction with a receptor. It is OK to suggest TGR5. It is not clear the “high-affinity interation”.

c) page 11, line 290: “… our results are consistent with a previous study…”. Please provide reference!

d) Page 11, line 299: “… it is unlikely that LCA inhibits either of these cytoplasmic proteins…”. Why is it unlikely? Please explore and clearly explain the scientific support for your hypotheses.

Additional comments

This paper has merit however it still presents several pitfalls. Some suggestions were reported as followed by the authors reply but the revised paper still presents several of those problems. Please correct carefully the paper!

---

## Round 0.3 · accepted · Accept

No further comments concerning the scientific content.